

# Neutrophil extracellular trap burden correlates with the stenosis of coronary atherosclerosis

Yan Wang[*], Mao Yang[*], Yuanyuan Xu, Shujun Yan, Enze Jin and Xueqi Li

Department of Cardiovascular Medicine, the Fourth Affiliated Hospital of Harbin Medical University, Harbin, China
[*] These authors contributed equally to this work.

## ABSTRACT

**Background**. Coronary atherosclerosis diseases (CADs) are associated with chronic inflammation. Neutrophil extracellular traps (NETs) are a type of novel proinflammatory cytokines whose levels are dramatically elevated in acute coronary syndrome. We conducted this study to further evaluate the association between circulating NET-associated markers and CAD in Chinese adults.

**Methods**. A total of 174 patients with CAD and 55 healthy controls were screened using percutaneous coronary intervention or coronary computed tomography angiography. Blood lipid levels, blood glucose levels, and blood cell counts were determined using commercial kits. Serum levels of myeloperoxidase (MPO) and neutrophil elastase (NE) were measured using ELISA. Double-stranded DNA (dsDNA) in serum was quantified using the Quant-iT PicoGreen assay. We also compared the circulating NET levels with various parameters in the study subjects.

**Results**. The levels of serum NET markers, dsDNA, MPO, and NE, were significantly elevated in patients with CAD, particularly in the severe group, consistent with the increase in neutrophil counts. The levels of NET markers correlated with the risk factors of AS, increasing with the number of risk factors. NET markers were identified as independent risk factors for severe coronary stenosis and also as predictors of severe CAD.

**Conclusion**. NETs may be related to AS and serve as indicators or predictors of stenosis in patients with severe CAD.

Corresponding author
Xueqi Li, LixueqiLXQ@163.com

# INTRODUCTION

Coronary atherosclerosis disease (CAD) is not only prevalent in the elderly population, but remains a growing public health threat. There are many risk factors for atherosclerosis (AS), including diabetes, hypertension, hyperlipidemia, smoking, accumulation of lipids in the intima, and activation of the endothelium (*Thompson et al., 2013*). In addition, people with CAD have an increased susceptibility to develop myocardial infarction and ischemic stroke, which have been among the gravest diseases with morbidity and mortality

worldwide (*Glass & Witztum, 2001*). Although extensive research has been conducted to illustrate the pathogenesis of CAD, the underlying mechanisms remain largely unknown.

Fortunately, studies conducted in recent years have identified a link between inflammatory cytokines, such as C-reactive protein (CRP) and peroxisome proliferator-activated receptor (PPAR), and atherogenesis (*Kianoush et al., 2017*; *Li et al., 2016*; *Sabanoglu & Inanc, 2022*). It has been shown that recruitment of leukocytes and induction of proinflammatory cytokine production promote thrombosis (*Fuchs et al., 2010*; *Quan et al., 2020*). In addition, Net-mediated initiation of macrophages promotes the self-amplified IL-1-IL-17 cascade, while macrophages secrete the cytokine interleukin-1 $\beta$ (IL-1 $\beta$), leading to activation of the T helper 17 ($T_H17$) cells response, which further recruits more immune cells to the atherosclerotic lesions (*Warnatsch et al., 2015*). Moreover, stimulated T lymphocytes and mast cells secrete proinflammatory cytokines that contribute to local inflammation, proteolysis, and plaque rupture and growth (*Nauseef & Borregaard, 2014*). More studies have reported that neutrophils, as the first line of the initial immune response, are related to AS (*Soehnlein, 2012*). Activated neutrophils release neutrophil granule proteins, including myeloperoxidase (MPO), neutrophil elastase (NE), and cathepsin G, and chromatin into the extracellular space to form a fibril matrix known as neutrophil extracellular traps (NETs), resulting in a unique type of cell death known as NETosis (*Brinkmann et al., 2004*; *Wang et al., 2009*). A study has found that NETs can be formed not only in infectious conditions but also under inflammatory sterile states (*Qin et al., 2016*). However, in addition to their role in bacterial immunity, increasing evidence suggests that NETs contribute to the pathogenesis of autoimmune diseases, including rheumatoid arthritis (RA), systemic lupus erythematosus (SLE), and vasculitis (*Grayson & Kaplan, 2016*). In addition, NETs can also be found in tissues as well as inside blood vessels (*Meegan et al., 2017*). A study reported that neutrophil depletion in ApoE knockout mice significantly reduced lesion burden in atherogenesis (*Döring et al., 2012*). Recent research has also demonstrated high levels of neutrophil-derived enzymes in the serum of patients with acute coronary syndrome (*Mangold et al., 2015*). However, the role of these factors in NETs formed under clinical conditions is yet to be investigated. The objective of the present study was to analyze the relationship between serum NET markers and AS risk factors or the stricture degree of coronary arteries.

## MATERIALS & METHODS

### Ethics statement

All participants provided written informed consent before entering the study according to the Declaration of Helsinki. We also applied the license in the human research ethics committee of the Fourth Affiliated Hospital of Harbin Medical University (approval number 2016-LLSC-120).

### Participants

The research group included 174 adult patients, suspected to have CAD, undergoing percutaneous coronary intervention (PCI), who were admitted to the cardiology department of Fourth Affiliated Hospital of Harbin Medical University from April 2016

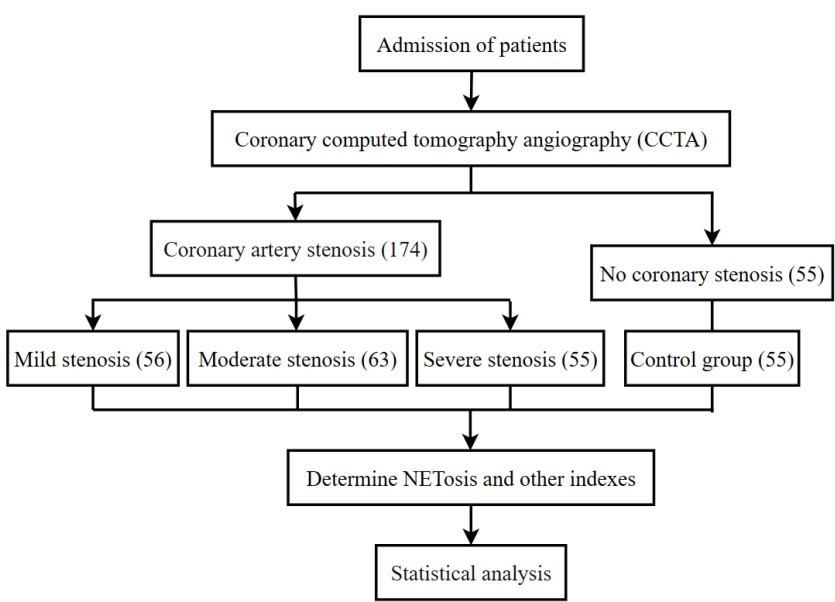

**Figure 1** Flowchart of the study selection procedure.

to December 2017. Patients with CAD free of severe chronic illness were divided into the following three groups: mild stenosis (<50%), moderate stenosis (50%–80%), and severe stenosis (>80%). Patients with advanced cancer, previous myocardial ischemia or infarction, and renal or liver disease were excluded. The control group included 55 subjects who were enrolled in the health examination center of Fourth Affiliated Hospital of Harbin Medical University, and all subjects were confirmed to have no luminal stenosis by coronary computed tomography angiography (CCTA). The whole study selection procedure is shown in Fig. 1. The control subjects must satisfy the following conditions: (1) no risk factors for AS, such as diabetes, hypercholesterolemia, and hypertension, and (2) negative findings on ECG. In addition, the following eight risk factors for coronary atherosclerosis were considered in all subjects: (1) BMI exceeding 24 kg/m$^2$, (2) systolic pressure exceeding 130 mmHg or diastolic pressure exceeding 85 mmHg, (3) blood glucose level exceeding 6.0 mmol/L, (4) plasma triglyceride (TG) level exceeding 1.69 mmol/L, (5) plasma cholesterol level exceeding 5.17 mmol/L, (6) high-density lipoprotein cholesterol (HDL-C) level less than 1.02 mmol/L, (7) low-density lipoprotein cholesterol (LDL-C) level exceeding 3.12 mmol/L, and (8) a current smoking history of more than one year.

## Blood sample collection and testing

Venous blood was collected aseptically from each patient and control subject into pyrogen-free tubes. Plasma glucose levels were immediately measured using the glucose oxidase method. The WBC count, neutrophil count, and platelet count were determined using an automatic hematology analyzer (SysmexXN-B3; Sysmex, Kobe, Japan). The levels of total cholesterol (TC), TG, HDL-C, and LDL-C were analyzed using an automatic biochemical analyzer (Hitachi 747; Hitachi, Tokyo, Japan). Venous blood was allowed to clot at room

temperature for at least 30 min, spun, and aliquoted, and the obtained serum was frozen and stored at $-80\,°C$ for the determination of NET-associated markers.

## Measurement of serum NET enzymes

The serum concentrations of NE and MPO were measured using ELISA kits (eBioscience, Vienna, Austria). Briefly, the serum samples were thawed at room temperature, diluted 50-fold in the supplied sample diluent, and then added to precoated ELISA plates. Assays were then performed according to the manufacturer's protocol. The absorbance of each microwell was read using a fluorescence spectrophotometer (Varian, Palo Alto, CA, USA) at the primary wavelength of 450 nm. The absorbance of both samples and human MPO or NE standards was determined. The concentrations of serum human MPO or NE were calculated according to the standard curve.

The serum level of double-stranded DNA (dsDNA) was quantified using the Quant-iT™ PicoGreen dsDNA reagent (P11496; Invitrogen, Carlsbad, CA, USA). Serum was diluted 10-fold in the working solution and incubated for 5 min at room temperature in the dark. The absorbance of samples was read at an excitation wavelength of 480 nm and an emission wavelength of 520 nm using a fluorescence spectrophotometer (Varian, California, USA). The fluorescence was calculated by subtracting the fluorescence value of the reagent blank from that of each of the samples. The concentration of dsDNA was determined using the dsDNA standard curve.

## Statistical analysis

Statistical analyses were conducted using the GraphPad Prism 5 or SPSS software package 19.0. Continuous data are expressed as mean ± standard deviation, and categorical variables are presented as numbers (percentages). The one-way analysis of variance test including Bonferroni correction was used to perform group comparisons. The relationship between NET-associated markers and AS risk factors was analyzed using Spearman correlation analyses. Multinomial logistic regression analyses were performed to identify the relationship between serum levels of NETs and the degree of luminal stenosis severity. Receiver operating characteristic (ROC) curve analysis was applied to determine the diagnostic value of NETs for the stenosis of CAD. $P$ values of $< 0.05$ were considered to be statistically significant.

# RESULTS

## Basic information of subjects

The control subjects had no risk factors for AS, such as obesity, smoking, diabetes, hypercholesterolemia, and hypertension, and therefore showed lower levels of all parameters than patients with CAD. Patients with CAD were divided into three cohorts according to the degree of stenosis as evaluated using PCI. As anticipated, there were no significant differences in age, gender, and smoking between the control and CAD groups. There was also no considerable difference in plasma HDL-C or TG levels among subjects with CAD. However, the levels of TC or LDL-C in plasma were significantly increased with the degree of stenosis. The severe group showed a higher prevalence of obesity (BMI ≥ 25

**Table 1  Major clinical characteristics of study subjects stratified by CAD severity.**

| Variable | Controls (n = 55) | Mild CAD (n = 56) | Moderate CAD (n = 63) | Severe CAD (n = 55) | P-value |
|---|---|---|---|---|---|
| Age (years) | 59.40 ± 13.79 | 57.07 ± 8.67 | 60.32 ± 9.73 | 61.60 ± 8.64 | 0.132 |
| Male/female | 26/29 | 26/30 | 35/28 | 25/30 | 0.597 |
| BMI (kg/m$^2$) | 24.60 ± 2.82 | 25.83 ± 3.22 | 27.63 ± 2.56 | 26.77 ± 4.03 | <0.001 |
| Smoking, n (%) | 8 (14.55%) | 8 (14.29%) | 15 (23.44) | 14 (25.45) | 0.248 |
| Hypertension, n (%) | 0 (0%) | 18 (32.14) | 27 (42.19%) | 31 (56.36) | <0.001 |
| Glucose (mmol/L) | 5.66 ± 1.13 | 5.67 ± 1.03 | 5.93 ± 1.78 | 6.84 ± 2.74 | <0.01 |
| CHO (mmol/L) | 4.35 ± 0.77 | 5.09 ± 0.83 | 5.16 ± 0.91 | 5.07 ± 1.18 | <0.001 |
| TG (mmol/L) | 1.54 ± 0.89 | 1.80 ± 0.92 | 1.86 ± 1.08 | 1.92 ± 1.02 | 0.188 |
| HDL-C (mmol/L) | 1.14 ± 0.28 | 1.19 ± 0.20 | 1.19 ± 0.24 | 1.19 ± 0.43 | 0.724 |
| LDL-C (mmol/L) | 2.52 ± 0.53 | 3.13 ± 0.71 | 3.20 ± 0.69 | 3.22 ± 1.03 | <0.001 |

**Notes.**

Patients were divided into control, mild, moderate and severe CAD based on the severity of coronary artery disease (CAD), continuous data are expressed as mean +/− standard deviation and categorical variables are presented as numbers (percentages). Statistical significance at the $P < 0.05$ level. The one-way analysis of variance test including Bonferroni correction were used to perform group comparisons.

CAD, coronary artery disease; BMI, Body mass index; HDL-C, high-density lipoprotein cholesterol; LDL-C, low-density lipoprotein cholesterol; CHO, cholesterol; TG: Triglyceride.

kg/m$^2$), hypertension, and diabetes mellitus than the mild group. Moreover, the levels of TC, LDL-C, and blood glucose were increased with an increase in the degree of stenosis in patients with CAD (Table 1).

## Serum levels of NET markers in patients with CAD

We evaluated the NET-associated biomarkers, including MPO, NE, and dsDNA, in the control and CAD cohorts using commercial ELISAs. Interestingly and as expected, the levels of NET-associated biomarkers were increased in patients with CAD compared with those in matched controls (Figs. 2A, 2B and 2C). Moreover, circulating NET levels were significantly higher in patients with CAD with severe stenosis than in patients with mild or moderate stenosis. As these markers are the products of WBCs or PMNs, we assumed that the abovementioned changes could be related to an increase in neutrophil numbers, which has been reported previously (*Fuchs et al., 2010*). Our results also demonstrated that the increase in WBC or PMN counts was most prominent in severe subjects, suggesting that the finding could be related to the degree of stenosis of CAD (Figs. 2C and 2D), whereas there were no significant differences in platelet counts among the groups (Fig. 2E).

## Correlation between NET markers and CAD risk factors

To understand the mechanisms of increase of NET-associated biomarkers in patients with CAD, we investigated the relationship between NET markers and CAD risk factors. Because neutrophils are the major source of NETs, we first observed their relationship (*Borissoff et al., 2013*). In all the study subjects, the levels of dsDNA, MPO, and NE significantly and positively intercorrelated with the counts of WBCs and PMNs, especially the level of dsDNA (Table 2). We next explored the association between serum NET-associated marker levels and AS risk factors using a nonparametric correlation analysis of variables in all study subjects. As shown in Table 2, the serum MPO level correlated positively with hypertension

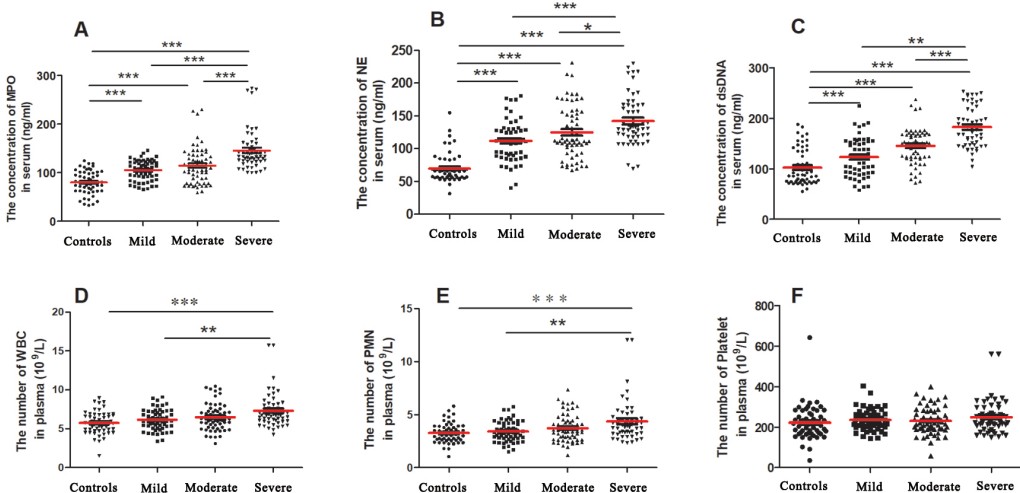

**Figure 2 Serum levels of NET-associated parameters in patients with CAD and controls.** Serum levels of MPO, NE, and dsDNA in healthy controls and all patients with CAD (A, B, C). Total counts of WBCs, PMNs, and platelets in plasma in healthy controls and all patients with CAD (D, E, F). Statistical significance is indicated by asterisks (* $P < 0.05$, ** $P < 0.01$, *** $P < 0.001$). The one-way analysis of variance test including Bonferroni correction was used to perform group comparisons. dsDNA, double-stranded DNA; MPO, myeloperoxidase; NE, neutrophil elastase; PMN, polymorphonuclear.

($r_s = 0.258$, $P < 0.001$) and hyperlipidemia (CHO, TG, and LDL-C; all $P < 0.01$). However, the MPO level did not correlate with BMI, smoking, and blood glucose levels. Furthermore, circulating dsDNA and NE levels correlated positively with BMI, CHO and LDL-C levels (all $P < 0.05$). A nonsignificant correlation was observed between dsDNA levels and TG, HDL-C, smoking, and blood glucose levels. The dsDNA levels also showed no close correlation with smoking and HDL-C levels. In Table 3, although the association between these NET-associated biomarkers and AS risk factors was weak (all $R < 0.5$), the effect was very significant ($P < 0.001$). Next, we analyzed the relationship between the levels of circulating NETs and the number of components of risk factors in one subject (Fig. 3). We observed that the levels of circulating NETs, in all study subjects, increased progressively with an increase in the number of types of risk factor components. Moreover, as can be seen from Fig. 3, with the concentration of NE in serum increased, there was a significant positive correlation with the number of types of risk factor component ($P < 0.01$).

## Relationship between CAD severity and risk factors and diagnostic value of NETs

We investigated the association between the risk factors and degree of stenosis of CAD using multinomial logistic regression analyses (Table 4). We detected increasing concentrations of serum NE in all patients with CAD, including those with mild CAD, moderate CAD, and severe CAD. In particular, severe CAD was significantly and positively associated with all NET biomarkers. LDL-C levels also independently predicted the presence of severe

**Table 2    Relationship between NET-associated markers and CAD risk factors.**

| Variable | MPO | | NE | | dsDNA | |
|---|---|---|---|---|---|---|
| | $r_s$ | $P$-value | $r_s$ | $P$-value | $r_s$ | $P$-value |
| BMI | 0.096 | 0.148 | 0.262 | <0.001 | 0.182 | <0.01 |
| Smoking | 0.017 | 0.794 | 0.073 | 0.273 | 0.101 | 0.129 |
| Hypertension | 0.258 | <0.001 | 0.350 | <0.001 | 0.356 | <0.001 |
| Glucose | 0.057 | 0.391 | 0.041 | 0.537 | 0.146 | <0.05 |
| CHO | 0.315 | <0.001 | 0.264 | <0.001 | 0.152 | <0.05 |
| TG | 0.186 | <0.01 | 0.127 | 0.054 | 0.187 | <0.01 |
| HDL-C | 0.136 | <0.05 | 0.061 | 0.357 | 0.033 | 0.616 |
| LDL-C | 0.249 | <0.001 | 0.193 | <0.01 | 0.149 | <0.05 |
| WBC | 0.210 | <0.01 | 0.157 | <0.05 | 0.413 | <0.001 |
| PMN | 0.206 | <0.01 | 0.145 | <0.05 | 0.364 | <0.001 |

**Notes.**

Nonparametric correlation analysis of variables associated with circulating NETs levels in all the study population.
$r_s$, Spearman rho.

**Table 3    Relationship between NET-associated markers and number of risk factors.**

| Variable | Number of risk factors | | | |
|---|---|---|---|---|
| | R | $R^2$ | F | $P$-value |
| MPO | 0.262 | 0.069 | 16.779 | $P < 0.001$ |
| NE | 0.29 | 0.084 | 20.857 | $P < 0.001$ |
| dsDNA | 0.312 | 0.097 | 24.418 | $P < 0.001$ |

**Notes.**

Correlation analysis between the number of risk factors and NET related biomarkers.

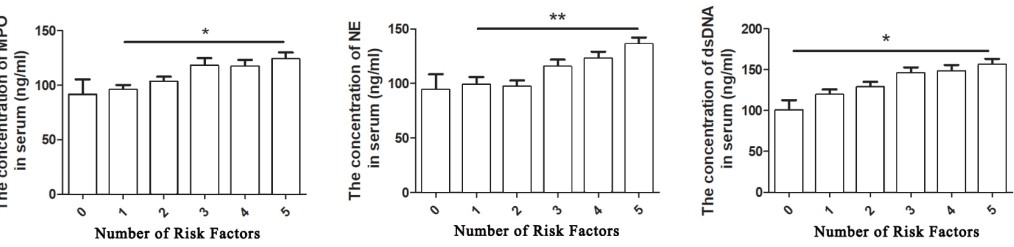

**Figure 3    Relationship between NET-associated markers and risk factors for coronary atherosclerosis in all subjects.** The one-way analysis of variance test including Bonferroni correction was used to perform group comparisons. Statistical significance is indicated by asterisks (* $P < 0.05$, ** $P < 0.01$, *** $P < 0.001$); 0, 1, 2, 3, 4, 5 indicate the number of types of risk factors in one subject.

stenosis. There were no significant differences between mild CAD and moderate CAD with respect to other risk markers of CAD.

Furthermore, we performed ROC curve analysis and evaluated the diagnostic accuracy and cutoff values of MPO, NE, and dsDNA to determine the diagnostic value of NET markers for the stenosis of CAD (Fig. 4). The best cutoff value of plasma MPO to predict

**Table 4  Multinomial logistic regression models for CAD severity.**

| Variable | Mild CAD | | | Moderate CAD | | | Severe CAD | | |
|---|---|---|---|---|---|---|---|---|---|
| | OR | 95% CI | *P*-value | OR | 95% CI | *P*-value | OR | 95% CI | *P*-value |
| Age | 0.99 | (0.94–1.04) | 0.60 | 1.03 | (0.97–1.08) | 0.40 | 1.03 | (0.96–1.10) | 0.48 |
| BMI | 1.04 | (0.85–1.28) | 0.72 | 1.24 | (1.00–1.54) | 0.05 | 1.11 | (0.87–1.42) | 0.40 |
| Glucose | 1.05 | (0.68–1.64) | 0.82 | 1.16 | (0.75–1.82) | 0.50 | 1.66 | (1.02–2.68) | 0.04 |
| CHO | 1.26 | (0.47–3.43) | 0.65 | 1.22 | (0.44–3.40) | 0.70 | 0.86 | (0.27–2.72) | 0.80 |
| TG | 0.91 | (0.43–1.91) | 0.80 | 0.90 | (0.42–1.93) | 0.79 | 0.70 | (0.30–1.64) | 0.42 |
| HDL-C | 1.47 | (0.09–23.23) | 0.79 | 1.20 | (0.07–21.96) | 0.90 | 0.12 | (0.00–3.40) | 0.21 |
| LDL-C | 3.88 | (1.15–13.10) | 0.03 | 4.91 | (1.41–17.13) | 0.01 | 5.13 | (1.22–21.54) | 0.03 |
| WBC | 2.12 | (0.92–4.92) | 0.08 | 1.62 | (0.66–3.98) | 0.30 | 1.96 | (0.70–5.52) | 0.20 |
| PMN | 0.48 | (0.16–1.42) | 0.19 | 0.75 | (0.24–2.38) | 0.63 | 0.60 | (0.16–2.23) | 0.45 |
| Platelet | 1.01 | (1.00–1.02) | 0.27 | 1.01 | (0.99–1.02) | 0.37 | 1.01 | (0.99–1.02) | 0.35 |
| MPO | 1.02 | (0.99–1.04) | 0.29 | 1.02 | (1.00–1.05) | 0.10 | 1.06 | (1.03–1.10) | 0.00 |
| NE | 1.05 | (1.02–1.07) | 0.00 | 1.05 | (1.02–1.07) | 0.00 | 1.04 | (1.02–1.07) | 0.00 |
| dsDNA | 1.00 | (0.98–1.02) | 0.81 | 1.01 | (0.99–1.03) | 0.23 | 1.04 | (1.02–1.07) | 0.00 |
| Gender | 0.60 | (0.16–2.23) | 0.45 | 0.44 | (0.11–1.80) | 0.25 | 0.77 | (0.15–4.00) | 0.76 |
| Smoking | 0.70 | (0.13–3.75) | 0.68 | 0.60 | (0.11–3.31) | 0.56 | 0.46 | (0.07–3.06) | 0.42 |

**Notes.**

Statistical significance at the *P* < 0.05.

severe CAD was 123 ng/ml (sensitivity 78%, specificity 22%, and AUC 0.85), that of plasma NE to predict severe CAD was 116 ng/ml (sensitivity 78%, specificity 29%, and AUC 0.77), and the best cutoff value of plasma dsDNA to predict severe CAD was 126 ng/ml (sensitivity 90%, specificity 44%, and AUC 0.75).

## DISCUSSION

To our knowledge, this is the first study to investigate the correlation between NET markers and risk factors or severity of stenosis of CAD in Chinese adults. Our study demonstrated that the levels of serum NET markers were significantly elevated with the degree of coronary artery stenosis, particularly in subjects with severe CAD, and correlated with risk factors for AS, increasing with the number of risk factors. These results suggest that NETs are related to AS and serve as indicators or predictors of stenosis in patients with severe CAD.

Atherosclerosis is a chronic inflammatory disease of the arterial wall with localized plaque formation (*Farmer & Torre-Amione, 2002*). *Jia et al. (2017)* mentioned anti-thrombotic therapy for plaque erosion in their study, and this work is also helpful for our study on the rationale of NETosis, which can more accurately screen plaque types, and more targeted and reliable results for the study of neutrophil trapping nets. Although the exact cause remains unknown, some risk factors have been established to be involved in the occurrence and development of atherosclerosis (*Li et al., 2016*). In this study, we evaluated the relationship between conventional risk factors, including diabetes, dyslipidemia, tobacco smoking, abdominal obesity, hypertension, age, gender, and stenosis of CAD. Various degrees artery stenosis has no significant relationship with sex, age, smoking, and HDL-C. A common explanation is that the scope of PCI or CCTA service is relatively small and the admitted

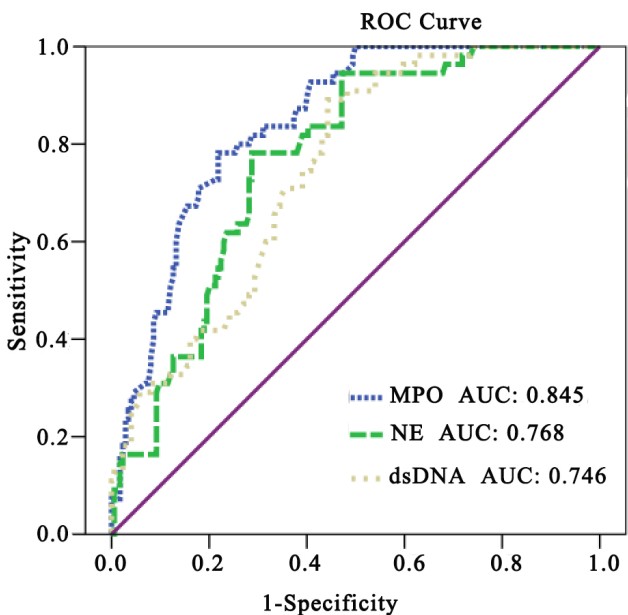

**Figure 4** **RUC curve analysis was used to evaluate the diagnostic accuracy of Net-associated markers in predicting the stenosis severity of CAD.** Values are mean ± SD.

populations are older in age. Extensive research has shown that conventional cardiovascular risk factors are incompletely predictive of CAD, particularly in elderly patients, suggesting the need to develop alternative approaches (*Kats et al., 2016*).

Atherosclerosis-induced injury of arterial wall is related to lipid infiltration, autophagy activation, reactive oxygen species (ROS) generation, and metabolic stress (*Alves & Ames, 2003*). Increasing evidence has shown that neutrophils or NETs have key roles in the onset of CAD complications, especially stenosis or calcification (*Borissoff et al., 2013*). In contrast to the findings of *Qin et al. (2016)*, our study demonstrated that the levels of NET-associated markers increased according to the degree of stenosis, a relationship that is linearly proportional to the counts of leukocytes or neutrophils. We speculate that this is due to the different disease state of the study subjects. The patients studied by *Qin et al. (2016)* were type 1 diabetes mellitus (T1D) with a single disease, while the patients observed in this study tend to present with multiple diseases or risk factors. Moreover, the difference in experimental conditions may lead to different levels of NETs. In Jim Qin's study, serum samples were diluted 100-fold in the supplied buffer provided to determine the serum levels of PR3 and NE. However, in this study, serum levels of NE and MPO were measured by dilution of 50-fold.

While focusing on these results, we were more interested in what induced the increased levels of NET-associated markers during the CAD. Therefore, we next analyzed the relationship between NET-associated markers and AS risk factors in all the study subjects. Although our results showed a weak association between NET-associated biomarkers and AS risk factors, but the effect was very significant. This may be because the sample size was too small, or the risk of per patient was not as high as in other published papers. In addition,

consistent with recently published data by Julian et al. on CAD in Europeans (*Borissoff et al., 2013*), we also confirmed that the levels of NETs were closely related to hypertension, high levels of cholesterol and LDL-C, and especially, the number of types of risk factors in one subject. We observed increasing levels of dsDNA with all types of risk factors, including obesity, hypertension, hyperlipidemia, and hyperglycemia. We speculate that the chronic stimulation of different risk factors, especially, hypertension, hyperlipidemia, and hyperglycemia, activates neutrophils and the formation of NETs. Both neutrophil recruitment and NET formation in blood vessel walls contribute to vascular occlusion by injuring the endothelium and inducing thrombosis (*Wang et al., 2018*; *Qi, Yang & Zhang, 2017*). Some studies also showed that neutrophils aggravate atherosclerosis by priming the inflammatory response of the macrophage, the major inflammatory cell in AS (*Drechsler et al., 2011*; *Wantha et al., 2013*).

## CONCLUSIONS

We explored whether NETs are mediators of atherosclerosis. Serum NET levels correlated positively with CAD risk factors and also with the degree of stenosis. Our data support the hypothesis that NETs can be considered as a marker of CAD and serve as a better predictor for diagnosing the stenosis of CAD.

Our study also has some limitations, and a further animal/cell study and clinical intervention study on NETs are necessary in the future to provide more valuable support for clinical application. All samples were derived from Chinese subjects, due to which the results cannot be generalized. In the future, we intend to analyze the serial changes of NETs in circulation in different stages of CAD.

### Funding
The authors received no funding for this work.

### Competing Interests
The authors declare there are no competing interests.

### Author Contributions
- Yan Wang performed the experiments, analyzed the data, prepared figures and/or tables, authored or reviewed drafts of the article, and approved the final draft.
- Mao Yang performed the experiments, analyzed the data, prepared figures and/or tables, and approved the final draft.
- Yuanyuan Xu performed the experiments, analyzed the data, prepared figures and/or tables, and approved the final draft.
- Shujun Yan performed the experiments, analyzed the data, prepared figures and/or tables, and approved the final draft.
- Enze Jin performed the experiments, analyzed the data, prepared figures and/or tables, and approved the final draft.

- Xueqi Li conceived and designed the experiments, authored or reviewed drafts of the article, and approved the final draft.

## Clinical Trial Ethics

The following information was supplied relating to ethical approvals (*i.e.*, approving body and any reference numbers):

The study was approved by the Ethics Committee of the Fourth Affiliated Hospital of Harbin Medical University.

## Data Availability

The raw data are available in the Supplementary File.

## Supplemental Information

Supplemental information for this article can be found online at http://dx.doi.org/10.7717/peerj.15471#supplemental-information.

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
