# Peer review of "Neutrophil extracellular trap burden correlates with the stenosis of coronary atherosclerosis"

_PeerJ, doi:10.7717/peerj.15471_

## Round 0.1 · original submission · Minor Revisions

The manuscript related to the study of the presence of NETs in serum as a biomarker for AS risk has been revised and minor revisions are required. Please, follow the reviewers' suggestions.

Reviewer 1 ·

Basic reporting

The presented manuscript aimed to study NETs presence in serum as a biomarker for AS risk. The results from the study highlight a positive correlation between serum NET levels with CAD risk and with the degree of stenosis in Chinese adult patients with the healthy control group.

Experimental design

Please add following corrections to the manuscript-

NETs are not cytokines, please correct the statement in the background and the whole text. See the literature for reference (Blood (2019) 133 (20): 2178–2185).
• Line 54- 57, please elaborate it clearly. The lines start abruptly and lack full information on how NETs activate macrophages and T cells (mechanistically).
• The introduction part must be restructured for a clear information flow and a better understanding of the study rationale.
• Add a flow chart for the study selection procedure for clarity.
• Line 160, add the reference.
• Line 157-173- the description of the results is ambiguous and not referenced to figure 2 panels, it is hard to follow the data. Add reference of correct figure panel to corresponding results description, unlike current format where the figure 2 reference is given once in the end.
• Table 2 shows NETs biomarkers and AS risk factors correlation but Weakley ( all below rs value 0.5). Please mention this in the results section and add some explanation in the discussion part.
• Figure 2, graph the data in linear regression with R2 values for a better correlation conclusion.
• Figure 3. Add labels to all lines. And describe the figure details in legends.

Validity of the findings

The data clearly supports that NET-associated biomarkers in circulation suggest potential biomarkers for CAD severity. The outcome of this study will certainly add to current knowledge and understanding of AS pathogenesis and prognosis. However, the lack of supporting data from in vivo animal studies where blocking neutrophils in AS disease model raises the concern for realistic use of these results in clinics.

Reviewer 2 ·

Basic reporting

i) I recommend re-phrasing lines 216-217 since it jumps suddenly into the present tense, and it is unclear what that sentence is meant to convey.

ii) The graph legends in figure 1 require consistent punctuation in that the titles of all treatment conditions must begin with letters in the upper case e.g. “Mild,” and not “mild.”

iii) The graph legends in Figure 2 will prove more informative if changed to “Number of Risk Factors”. Plus, I think including correlational analyses (PCA or linear regressions) for a few risk factors will demonstrate the relationship between risk factors and NET-associated biomarkers.

iv) The EROSION study by Jia et al (European Heart Journal, Volume 38, Issue 11, 14 March 2017, Pages 792–800) may not be a direct examination of the role of NETosis in cardiovascular disease, they do talk about anti-thrombotic therapy for plaque erosion. Commenting on this work will also strengthen the authors’ rationale for examining NETosis. Incidentally, the EROSION study is also a product of The Harbin Hospital.

v) In commenting on the Qin study (lines 209-215), it would be useful if the authors could briefly expand on the “different state of the study subjects” and “difference in experimental conditions.” I think doing so will further empower their results and conclusions.

Experimental design

I have no major grouses with the experimental design herein. My suggestions have been stated above.

Validity of the findings

No comment.

Additional comments

No comment.

---

## Round 0.2 · accepted · Accept

The revisions have been performed properly and I would like to thank the authors for their efforts.

Reviewer 2 ·

Basic reporting

The authors have made the requisite changes to their manuscript, and I am satisfied with the state of the paper as it is now.

Experimental design

The authors have made the requisite changes to their manuscript, and I am satisfied with the state of the paper as it is now.

Validity of the findings

The authors have made the requisite changes to their manuscript, and I am satisfied with the state of the paper as it is now.

Additional comments

The authors have made the requisite changes to their manuscript, and I am satisfied with the state of the paper as it is now.